# Linking Pharmacogenomic Information on Drug Safety and Efficacy with Ethnic Minority Populations

**DOI:** 10.3390/pharmaceutics12111021

**Published:** 2020-10-25

**Authors:** Dan Li, April Hui Xie, Zhichao Liu, Dongying Li, Baitang Ning, Shraddha Thakkar, Weida Tong, Joshua Xu

**Affiliations:** 1Division of Bioinformatics and Biostatistics, National Center for Toxicological Research, U.S. Food and Drug Administration, Jefferson, AR 72079, USA; dan.li@fda.hhs.gov (D.L.); april.xie@gmail.com (A.H.X.); zhichao.liu@fda.hhs.gov (Z.L.); Dongying.Li@fda.hhs.gov (D.L.); Baitang.Ning@fda.hhs.gov (B.N.); Shraddha.Thakkar@fda.hhs.gov (S.T.); Weida.Tong@fda.hhs.gov (W.T.); 2School of Pharmacy, Virginia Commonwealth University, Richmond, VA 23298, USA; 3Office of Computational Sciences, Office of Translational Sciences, Center for Drug Evaluation and Research, U.S. Food and Drug Administration, Silver Spring, MD 20993, USA

**Keywords:** pharmacogenomic, minority, data collection, drug, biomarker

## Abstract

Numerous prescription drugs’ labeling contains pharmacogenomic (PGx) information to aid health providers and patients in the safe and effective use of drugs. However, clinical studies for such PGx biomarkers and related drug doses are generally not conducted in diverse ethnic populations. Thus, it is urgently important to incorporate PGx information with genetic characteristics of racial and ethnic minority populations and utilize it to promote minority health. In this project a bioinformatics approach was developed to enhance the collection of PGx information related to ethnic minorities to pave the way toward understanding the population-wide utility of PGx information. To address this challenge, we first gathered PGx information from drug labels. Second, we extracted data on the allele frequency information of genetic variants in ethnic minority groups from public resources. Then, we collected published research articles on PGx biomarkers and related drugs for reference. Finally, the data were integrated and formatted to build a new PGx database containing information on known drugs and biomarkers for ethnic minority groups. This database provides scientific information needed to evaluate available PGx information to enhance drug dose selection and drug safety for ethnic minority populations.

## 1. Introduction

Pharmacogenomic (PGx) information can be utilized to improve the medical decision-making process and avoid severe adverse drug reactions. Drug labeling with PGx information can be used to promote drug safety and drug efficacy. In general, PGx studies include evaluation of genetic or genomic variations that serve as predictive PGx biomarkers to distinguish between responders and non-responders to specific drugs [1]. Knowing whether a patient carries a specific genetic variation or has altered expression levels of genes can help prescribers to individualize drug therapy, mitigate the chance of adverse events, and optimize the effectiveness of the drug with a proper dose [2,3,4,5]. Tools such as genotyping technologies and databases have been developed to incorporate the genetic information of patients into routine clinical practice to carefully identify and prescribe a drug to the appropriate ethnic group of patients [6,7]. However, a genetic variant associated with drug responses may show a widely diverse prevalence among populations, affecting drug efficiency and drug safety for different ethnic groups. Diverse ethnicities usually harbor variability in drug dose requirements, and genotyping of clinically relevant PGx biomarkers has been advocated to guide drug dosing/indication optimization for different patient populations [8]. Therefore, a reliable collection of PGx information with biomarkers/drugs and their behaviors in various minority groups is needed to provide guidance for effective and safe medication to treat ethnic minority populations. Unfortunately, clinical studies to characterize PGx biomarkers generally have not been conducted for diverse ethnic populations. Although some studies have considered the allele frequencies of genetic variants in different ethnic groups and have utilized genotype-guided algorithms for dose selection in the prescription of some drugs such as warfarin, acenocoumarol, and phencopromon to benefit certain ethnic populations [9,10], it remains challenging to improve the health of such groups with the current level of PGx information. Further systematic collection and evaluation of PGx biomarkers and related drugs for diverse ethnic groups is needed.

In this manuscript we describe the development and implementation of a bioinformatics workflow aimed to enhance the collection and utilization of PGx information related to ethnic minority groups (Figure 1). Labeling of a number of drugs and biologics approved by the United States Food and Drug Administration (FDA) include PGx information [11,12]; we utilized the information in the FDA Table of Pharmacogenomic Biomarkers in Drug Labeling (TPGxBMDL) from the FDA website (https://www.fda.gov/drugs/science-and-research-drugs/table-pharmacogenomic-biomarkers-drug-labeling). We also searched for PGx information containing relationships between drugs and their associated genetic variants from public resources such as DrugBank [13], Pharmacogenomics Knowledge Base (PharmGKB) [14], Clinical Interpretation of Variants in Cancer (CIViC) [15], and Gene Drug Knowledge Database (GDKD) [16]. Then, we filtered the collected data to focus on single nucleotide polymorphisms (SNPs) in the reported biomarkers that interact with non-oncology drugs. The allele frequency (AF) information of SNPs for different ethnicities were collected from the Allele Frequency Aggregator (ALFA) project [17], which provides a great opportunity to collect population-specific information for genetic variants. In total, allele frequency information of 67 SNPs associated with specific drug–biomarker pairs was collected. PGx drug and biomarker pairs were further explored in the PubMed database, and related articles with their PubMed IDs (PMIDs) were collected. Finally, all data were integrated into a database to provide researchers with information regarding drugs, paired PGx biomarkers, and associated AFs of SNPs for minority ethnic groups and to adjust drug doses and indications during drug development and drug application.

## 2. Methods

### 2.1. Data Collection and Processing

The therapeutic area for each drug along with its associated pharmacogenomic biomarkers was archived from TPGxBMDL. Drugs in the oncology therapeutic area were excluded from this study. Remaining non-oncology drugs were searched in four drug databases (DrugBank, PharmGKB, CIViC, and GDKD) to obtain drug–SNP interactions. We then normalized results by removing redundant information and built unique drug variant–gene interaction relationships.

The SNP allele frequency data was downloaded from ALFA. Only six ethnicities were investigated in this study due to an insufficient number of subjects from other ethnic groups (Appendix A). We focused on nonsynonymous SNPs in the coding region, and a further SNP filter was applied to restrict the SNPs to NSF, NSM, and NSN only. NSF, NSM, and NSN are explained below. Annotation for the SNPs was downloaded from the Single Nucleotide Polymorphism Database (dbSNP) developed and hosted by the National Center for Biotechnology Information (NCBI) (dbSNP human build 154: https://ftp.ncbi.nih.gov/snp/archive/b154/VCF/). Allele frequency was calculated as alternative allele counts divided by total allele counts.

NSF: Nonsynonymous frameshift. A coding region variation where one allele in the set changes all downstream amino acids.NSM: Nonsynonymous missense. A coding region variation where one allele in the set changes a protein peptide.NSN: Nonsynonymous nonsense. A coding region variation where one allele in the set changes to a stop codon, i.e., a termination codon.

### 2.2. Allele Frequency Thresholds Based on the European Population

The allele frequency of a SNP in the European group (*AF_e_*) was used as the baseline for the evaluation of the SNP’s allele frequencies among other ethnicities. A threshold was configured to identify the substantial changes in SNP allele frequency between the European group and another group. Specifically, when *AF_e_* < 0.05 the threshold is *AF_e_* + 0.05, which indicates, if the SNP AF in another group is greater than *AF_e_* by 0.05, the change is considered substantial. When *AF_e_* ≥ 0.4, the threshold was *AF_e_* ± 0.2. For higher *AF_e_* values, 0.2 was high enough to be considered substantially different between ethnic groups.

For 0.05 < *AF_e_* < 0.4, we fitted a formula that restricted the *AF* threshold depending on *AF_e_*. The threshold increased along with the *AF_e_* and had a maximum value of 0.2.
$$AF_{Threshold}=AF_{e}\pm AF_{diff}$$
$$AF_{diff}=\frac{0.4}{\left(1+0.4/\left(AF_{e}+0.01\right)\right)}$$

### 2.3. Literature Screening Process

We used the R package easyPubMed [18] to search and download abstracts of interest from PubMed. We tracked each article published after the year 2000 that contained a given drug name and then downloaded the Year, PMID, ArticleTitle, and AbstractText. Next, abstracts containing paired biomarkers associated with a drug were retained. Finally, to narrow the PubMed articles down to ethnic minority-associated studies, we selected nine secondary keywords (African, Asian, Latin, European, Chinese, American, pharmacogenomic, metabolizer, minority, metabolism, dose, hypersensitivity, adverse reaction) to compile the final literature candidates. PMIDs were then listed for each of the drugs.

## 3. Results

### 3.1. Data Collection by Drugs

The 2 February 2020 version of TPGxBMDL was downloaded from https://www.fda.gov/drugs/science-and-research-drugs/table-pharmacogenomic-biomarkers-drug-labeling. Therapeutic products from Drugs@FDA with drug labels containing PGx information are listed in this table. In total, 404 PGx drug–biomarker pairs were constructed with 283 unique drugs and 86 predetermined biomarkers (genes), excluding three fusion genes (PML-RARA, BCR-ABL1, and FIP1L1-PDGFRA). The therapeutic area and labeling sections of a given drug were included as well. As shown in Figure 2a, most of these drugs are in the therapeutic areas of Oncology (94), Psychiatry (34), and Infectious Diseases (31). In general, the biomarkers for oncology drugs are not related to germline variants. In most cases somatic mutations of cancer patients are investigated to select oncology drugs guided by the somatic mutation biomarkers. Therefore, we excluded the oncology drugs from our study to focus on other drugs to which ethnic minority populations may show a varied response.

To better present PGx information for various ethnic populations, we collected and investigated the allele frequency distribution of genetic variants associated with drugs and related genes across ethnic groups. We focused on six groups: African, South Asian, African American, Latin American, European, and Other, excluding groups in which studies contained fewer than 400 subjects per group (Appendix A). Drugs in FDA labels were searched and data on clinically actionable drug–gene interaction were collected from drug knowledge databases. Seventy-seven non-oncology drugs listed in TPGxBMDL were found in PharmGKB (75 drugs) and DrugBank (36 drugs). Most (34 of 36) of the drugs found in DrugBank were also included in PharmGKB (Figure 2b).

These 77 drugs were associated with 839 SNPs, as reported by PharmGKB and DrugBank. However, only 168 of these SNPs are located in 19 genes which are also listed as PGx biomarkers in TPGxBMDL. Allele frequency data for different ethnicities were then collected from the ALFA project, which contains approximately 447 million SNPs from more than 100,000 subjects of different ethnicities according to the newly released version 20200227123210. The ALFA dataset is updated quarterly with 100,000–200,000 new subjects of genotype and phenotype data, serving as a comprehensive and relevant reference [17]. Nonsynonymous SNPs that cause frameshift, missense, or nonsense, with ethnicity-specific allele frequency information were further selected (see Methods for details). As a result, only 67 SNPs with allele frequency demographic information were collected, which were related to 148 drug–SNP pairs made from 42 non-oncology drugs and 16 (out of 86) PGx biomarkers.

Figure 2c shows the therapeutic area distribution of these 42 drugs with data on related SNP allele frequencies. Some of the drugs interact with multiple SNPs. For example, warfarin interacts with 19 SNPs (Figure 2d). As warfarin is the most commonly prescribed oral anticoagulant used to treat thromboembolic disease, its narrow therapeutic index requires close attention to the individual variability in patient response. Studies have shown the importance of predicting or optimizing dose requirements among different ethnicities [9,19,20].

After removing rare SNPs (sum *AF* < 0.01), allele frequencies of the remaining 33 SNPs were evaluated. A two-way classification of allele frequencies was applied to group ethnicities and SNPs together (Figure 3a). Even though African and African American ethnicities were listed as two independent groups by the ALFA project, their allele frequencies for the SNPs were similar to each other and were, thus, grouped together. Meanwhile, the Other group was classified together with the European group, indicating this group may be a population or a mixed population genetically close to the European group. Allele frequencies of these SNPs for South Asian and Latin American groups were different from the other four groups, highlighting the genotype variability among diverse ethnic groups. It is apparent that rs1135840 exhibits high allele frequencies close to 0.6 in the African group and the other three groups compared to below 0.2 in the South Asian and Latin American groups. Located in CYP2D6, this SNP is associated with clozapine that is a widely used drug for schizophrenia treatment. Some other SNPs such as rs8103142 showed the highest allele frequency in the South Asian group (Figure 3a), indicating that South Asian and Latin American groups may require dose justification for certain drugs in comparison to other ethnic populations.

SNP allele frequencies were usually different from one ethnic group to another. Taking the European group that has a large number of subjects as the baseline, we compared the allele frequency of each SNP across ethnic groups (Figure 3b). Thresholds for substantial allele frequency difference were chosen according to the corresponding baseline values in the European group (see Methods for details). Any allele frequency value above or below the thresholds is marked in purple on the plot. Because African and African American groups were genetically close, the African group was not assessed in Figure 3b. As a result, 4, 0, 5, and 8 SNPs were identified with considerably different allele frequencies in the African American, Other, South Asian, and Latin American groups, respectively. Given that a total of 33 SNPs was assessed, 24.2% (8/33) of them exhibited substantially different allele frequencies in the Latin American group than in the European group. Moreover, 79% of these SNPs (26 of 33) showed a coefficient of variation over 0.5 (Figure 3c), indicating considerable diversity in allele frequency among ethnic groups, which health care providers should regard as significant for determination of patient treatment.

### 3.2. PGx Information of Biomarkers

According to the information provided in TPGxBMDL, 52 biomarkers interacted with non-oncology drugs. In 23 of these biomarkers we then identified 231 and 21 SNPs from PharmGKB and DrugBank, respectively. Fifteen SNPs were included in both resources, leaving 29 PGx biomarkers. These results suggest that the functions of SNPs in those 29 remaining PGx biomarkers may interact with non-oncology drugs.

Some genes, especially those from the Cytochrome P450 family, contained more drug-related SNPs than others, which have attracted more attention for PGx research (Figure 4a). However, when counting drugs that interact with these SNPs, many of them were not included in TPGxBMDL (Figure 4b), suggesting that the potential PGx information for these drugs has not been included in drug labels. For example, there are only 12 SNPs in the gene CYP2D6 (Figure 4a) that is one of the most important enzymes for the metabolism of xenobiotics. These 12 SNPs were found to be associated with 46 different drugs reported by PharmGKB and DrugBank, and only 16 (Figure 4b) were paired with CYP2D6 according to TPGxBMDL.

Next, we collected SNPs with allele frequency data in different ethnic groups reported by the current version of the ALFA database. As a result, 62 SNPs with allele frequencies in 15 PGx biomarkers were found to be associated with the non-oncology drugs in TPGxBMDL (Appendix A). Results showed that SNP information for many drugs is not available for diverse populations. Some biomarkers were not studied extensively, and inter-ethnicity variability was not addressed, which was likely due to a lack of enrollment of ethnic minorities in these clinical studies.

### 3.3. Literature Screening to Provide PGx PubMed IDs

We queried the drugs in labeling data to collect information related to ethnicity. Ethnicity-specific information such as metabolizer status (poor metabolizer, intermediate metabolizer, and ultrarapid metabolizer) for 40 drugs were identified in South Asian and African American ethnic groups; only 20 of these drugs were found to be associated with SNPs from the ALFA project, suggesting that more ethnicity-involved pharmacogenomics studies are needed for a larger number of drugs.

To further interpret the PGx data for diverse ethnicities and to provide additional PGx information for ethnic minorities, we searched PubMed to identify potential articles related to PGx in ethnic populations. Specifically, we searched and downloaded abstracts from studies that focused on drugs in TPGxBMDL. If an abstract contained a paired biomarker with a given drug and at least one of the keywords focused on ethnic groups (see Methods for details), the PMID of the abstract was added to the database as a potentially relevant article. Currently, 1329 articles covering 120 unique PGx drug–biomarker pairs have been identified and stored in the database (Appendix A, Methods).

Abacavir, a nucleoside analog that works as a reverse transcriptase inhibitor, is widely used to treat HIV infection. Per our screening, the PGx pair abacavir/HLA-B was mentioned in a great number of articles (19) that contained at least one keyword in the abstracts. However, no SNP with allele frequency information was reported by the ALFA project that links abacavir and HLA-B. We then reviewed the full articles for additional information on the diversity of ethnic samples involved in the studies. For example, the article PMID 29921043 reported an association between abacavir and biomarker HLA-B*57:01, where the authors demonstrated an ethnicity-specific association between the SNP and carbamazepine hypersensitivity via a comparison of risks among different ethnic populations of Asians in Hong Kong, Thailand, Malaysia, India, Korea, and Japan [21]. The PMIDs linked to articles that contain potential drugs and associated risk alleles with ethnicity-specific information were also stored in the database. The literature research potentially provided additional PGx information relevant to ethnic minorities. Unfortunately, most of these studies used a small number of subjects from ethnic minority populations, again indicating the lack of clinical PGx studies with sufficient numbers of ethnic minority subjects.

### 3.4. Database Development

Finally, using Microsoft Access, we developed a database of PGx information related to ethnic minorities. The main data table was built using each drug as a key basis of information. Each drug was listed in the first column, followed by columns listing SNPs and drug-interacting biomarkers. Allele frequencies of SNPs in six ethnic groups and in the total population were also listed, along with therapeutic area, and public databases that provided resources of drug SNP interaction information. Hyperlinks to public databases for each individual drug were also provided in the data table. Literature screening results were provided in a separate data table where related PMIDs were listed for each drug. The Access database is published as Appendix A and future updated versions will be shared upon request.

## 4. Discussion

We extracted PGx information from the ALFA database and four pharmacological databases. Although other public resources, such as 1000 Genomes [22] and gnomAD [23], could be used to acquire information on allele frequencies, the ALFA database contains SNP information for more ethnicities, and the SNP information in the ALFA database was derived from a larger sample size. Meanwhile, the ALFA database is evolving and SNP information from over 100,000 subjects is added quarterly. In our opinion, using the ALFA database will keep the information collection consistent and will benefit our database with continuous updates.

To demonstrate the utility of our database, here, we briefly describe the following case study. SNP rs3892097 *(CYP2D6***4)* is one of the most common variant alleles (allele frequency of 20%) in Caucasian populations [24]. It is a common nonfunctional allele leading to poor drug metabolism, accounting for over 75% of poor metabolizer patients [25]. We found seven antidepressant drugs that were impacted by this SNP from Clinvar, which is a public archive that aggregates information about genomic variation and its relationship to human health [26]: Amitriptyline, clomipramine, trimipramine, nortriptyline, imipramine, desipramine, and doxepin [27]. All drugs were included in TPGxBMDL. However, only nortriptyline and desipramine have polymorphism information in their drug labels. Since the ALFA project currently reports few and biased samples for this SNP among minority groups (10,434 of 11,174 were in the European and Other groups), we instead used gnomAD data that have slightly different ethnic groups than ALFA and contains 178,714 samples to compare the allele frequency of rs3892097 among various population groups: European—0.176, African—0.076, East Asian—0.0039, Latin American—0.114, and Other—0.151. It is estimated in drug labels that approximately 7–10% of Caucasians were poor metabolizers of P450 2D6 drugs. In light of the frequencies seen with this SNP, there is possibly an underestimation of poor metabolizer prevalence in the labeling among the European group. Substantially lower allele frequencies were found in African and South Asian populations compared to European populations, highlighting inter-population variability and the need for more data to confirm allele frequencies among the East Asian population. Our results indicate that more research is needed to enhance the collection of PGx information to further improve our understanding and practice of drug safety and efficacy for minorities.

Through our literature research we found over 40 PGx studies that tested non-oncology drugs for diverse ethnicities. Only 15 of these studies enrolled more than 100 patients, and numbers of ethnic minority patients were usually limited, accounting for about 5–20% of total participants. Literature research resulted in a small increase in the number of drugs with SNP information for ethnic minorities. Apparently, there is a lack of clinical studies on PGx biomarkers for these minority groups. Further, details of the analytical validation of methods used to detect and measure the biomarkers may not be available. Therefore, it is challenging to integrate all findings to further enhance our understanding of PGx information for ethnic minority groups.

## 5. Conclusions

We report here the development of a PGx knowledge database focusing on drugs and biomarkers in FDA drug labels to provide essential PGx information related to ethnic minorities. Such information is urgently needed for the promotion of drug safety and efficacy for diverse ethnic groups. Information on allele frequencies of SNPs related to drugs and PGx biomarkers in different ethnic populations provided an informative resource for improved clinical practice, by which appropriate drug selection and dose optimization can be conducted for diverse patient groups. By comparing multiple populations with the European population, which represents the largest sample size in the ALFA project, several SNPs were identified to exhibit distinct allele frequencies for other ethnicities, suggesting that closer attention should be paid when the same drug is prescribed for different ethnicities in the practice of personalized treatment. A literature research focusing on ethnic minority groups provided additional PGx information and highlighted the lack of studies on PGx biomarkers for ethnic minorities. The database developed in this study provides scientific support for drug reviewers and researchers to assess available PGx information for different ethnicities, which could promote the practice of personalized medicine in ethnic minority groups.

## Figures and Tables

**Figure 1 pharmaceutics-12-01021-f001:**
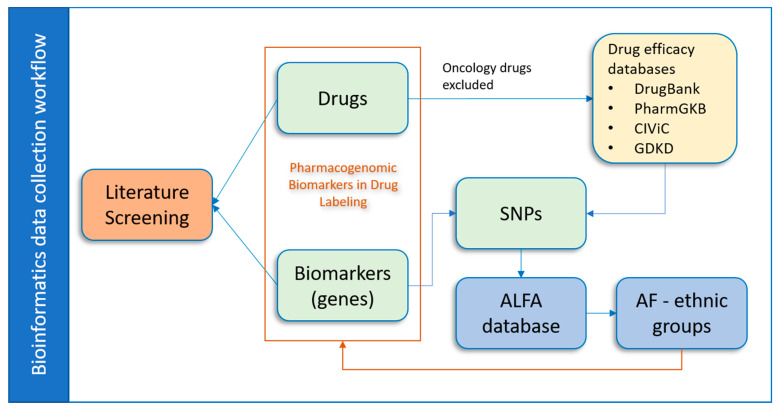
Overall data collection workflow. The paired PGx biomarkers and drugs were first downloaded from TPGxBMDL. The oncology drugs were excluded based on the therapeutic area information within TPGxBMDL. Then, SNPs that potentially interacted with the remaining drugs were collected from public drug efficacy databases. These SNPs thus linked the drugs and the PGx biomarkers (genes) in which they are located. Next, the AF information of the SNPs across ethnic groups were collected from the ALFA database. Additionally, PubMed literatures were searched for potential information on ethnic minority groups regarding the PGx biomarker and drug pairs.

**Figure 2 pharmaceutics-12-01021-f002:**
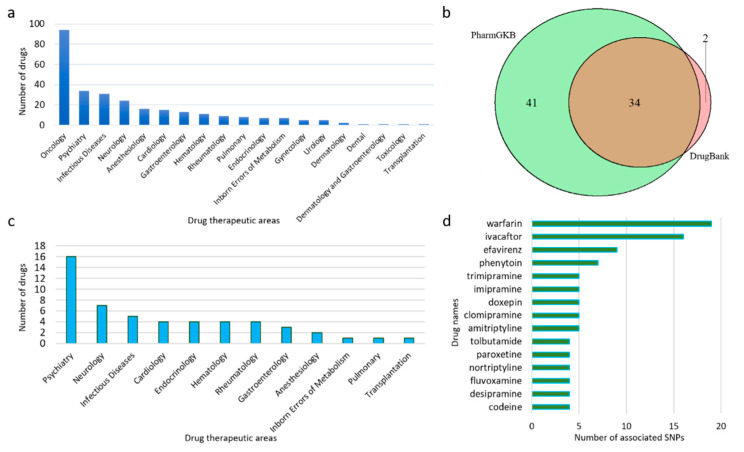
Summary of data collection by drugs. (**a**) The distribution of all drugs with PGx information in drug labels across therapeutic areas. (**b**) The numbers of non-oncology drugs with PGx information that were from PharmGKB and DrugBank. PharmGKB and DrugBank shared a great portion of drugs from our selection. (**c**) The distribution of therapeutic areas for non-oncology drugs with information on SNP allele frequency in six ethnic groups. (**d**) Top drugs that interact with the greatest number of SNPs with reported allele frequency information.

**Figure 3 pharmaceutics-12-01021-f003:**
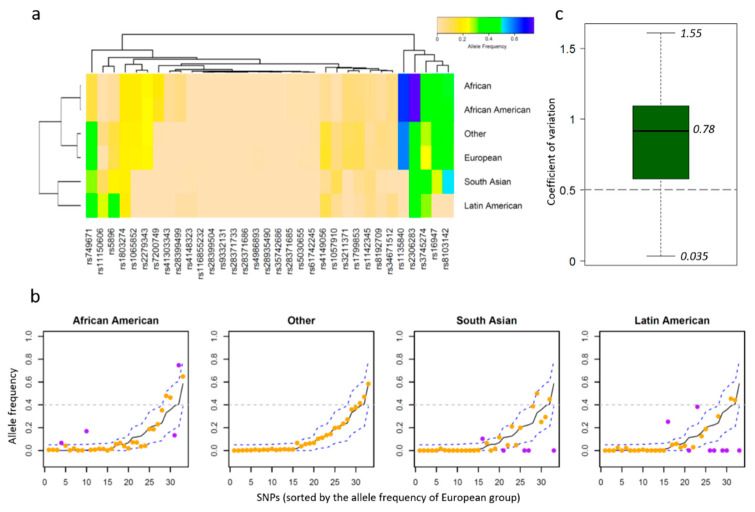
Analysis of the SNP allele frequency in diverse ethnic groups. (**a**) The heatmap of allele frequencies across ethnic groups (rows) and SNPs (columns). A two-way classification was performed on the allele frequency matrix. (**b**) The allele frequencies of the other four groups were plotted with those of the European population as baseline (middle black line). Any value beyond the thresholds (blue dash lines) is marked in purple, highlighting the substantial variability between ethnic groups. (**c**) The coefficient of variation distribution of SNP allele frequencies in six groups. The median coefficient of variation was 0.78, and 26 of 33 SNPs were with CVs over 0.5.

**Figure 4 pharmaceutics-12-01021-f004:**
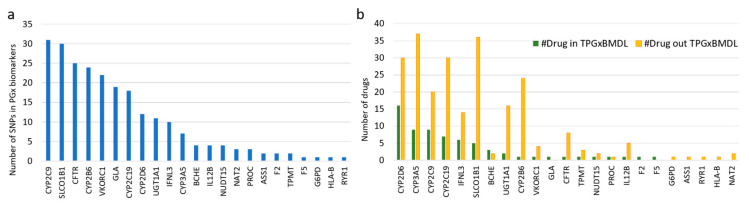
Numbers of SNPs and drugs related to biomarkers in TPGxBMDL. (**a**) The number of SNPs reported by public databases that are contained by biomarkers associated with drugs. (**b**) The number of drugs associated with SNPs in the biomarkers. Green bars represent the number of drugs listed in TPGxBMDL.

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
