# Peer review of "Linking Pharmacogenomic Information on Drug Safety and Efficacy with Ethnic Minority Populations"

_pharmaceutics, 2020, doi:10.3390/pharmaceutics12111021_

Round 1

Reviewer 1 Report

According to the guidelines for reviewers, I would like to report on my reviewing progress. Regarding the originality of this manuscript, the article is novel and interesting. The research question of the article is an important one. This article is in the top of 50% of papers in this field.

Author Response

Thank you very much for the comments.

Reviewer 2 Report

The manuscript "Linking Pharmacogenomic Information on Drug Safety and Efficacy with Ethnic Minority Populations" presents an interesting approach to data driven knowledge discovery utilizing a pharmacogenomic information related to ethnic minority groups. The manuscript is well written and properly structured. I would also note its condense content as an advantage. I assess high overall score. However, there are some issues which should be addressed by the authors.

1) As Pharmaceutics is the OA Journal, please attach as supplementary material the Access database derived from the literature mentioned in section 2.4.

2) Those seven case study drugs (API) are all tricyclic antidepressants (TCA), therefore it seems they all undergo the same metabolic pathway. We can assume that other TCAs would be affected as well. Do the authors have other case studies which would show the benefits of their database?

Concluding, I recommend the publication of reviewed manuscript in Pharmaceutics after minor revision.

Author Response

First of all, we want to thank you  for the constructive comments. The manuscript has been revised to address the reviewers’ concerns. Please see our point-to-point response below for Reviewer 2.

Reviewer 2:

The manuscript "Linking Pharmacogenomic Information on Drug Safety and Efficacy with Ethnic Minority Populations" presents an interesting approach to data driven knowledge discovery utilizing a pharmacogenomic information related to ethnic minority groups. The manuscript is well written and properly structured. I would also note its condense content as an advantage. I assess high overall score. However, there are some issues which should be addressed by the authors.

We appreciated the constructive comments.

1) As Pharmaceutics is the OA Journal, please attach as supplementary material the Access database derived from the literature mentioned in section 2.4.

Thanks for the suggestion, we have the Access database to be published with the manuscript as a supplementary file. Future updated versions will be shared upon request.

2) Those seven case study drugs (API) are all tricyclic antidepressants (TCA), therefore it seems they all undergo the same metabolic pathway. We can assume that other TCAs would be affected as well. Do the authors have other case studies which would show the benefits of their database?

We appreciate the reviewer’s interest in other case studies. Due to the manuscript length limit, we have chosen to use just one case study as an example. With the database published as a supplementary file, the readers will be able to extract information or find more case studies.

Concluding, I recommend the publication of reviewed manuscript in Pharmaceutics after minor revision.

Reviewer 3 Report

Li et al. have come up with a very novel approach aimed at deciphering the complex landscape of patterns involved in pharmacogenomic (PGx) interactions. The authors have deployed data mining techniques to show that, despite the ethnical-specific clinical data are being scarce, indeed, conserved PGx patterns can be found between individual race groups. They have summarized these findings in a comprehensive database, including references to the the relevant literature sources, that should be publicly shared with the scientific audience. This study, the manuscript of which is extremely well written, provides one of the key cornerstones for us to enter the future era of personalized medicine.

1) Would it be possible to stratify the drug-SNP interactions further to incorporate even more personalized criteria such as the sex or age of the patient for each ethnic group? This data could be plotted in a new figure.

2) Please change "Correspondence:Corresponding author:" to "Corresponding author:" or "Correspondence:" (line 12).

3) "pharmacogenomics" could be shortened to "PGx" (line 57).

4) "Pharmacogenomic" could be shortened to "PGx" (line 58).

5) Although the authors state that the PGx ethnic database will be provided to researchers "Finally, all data was integrated into a database to provide researchers with information regarding drugs, paired PGx biomarkers, and associated allele frequency of SNPs for minority ethnic groups and to adjust drug doses and indications during drug development and drug application" (line 69), it seems not to be publicly accessible "The database will be shared upon request" (line 222). Please either stress this fact in the former sentence or make the database open to everyone.

6) Would the authors mind providing a brief walk-through for Figure 1 in its legend? Please describe the process associated with each arrow. For example, it is not immediately clear what does "AF" mean and why does it connect to "Drugs" and "Biomarkers (genes)"?

7) Would it please be possible to have the 4 genes excluded from the drug-biomarker data set specified in parentheses (line 82)?

8) Would it please be possible to have the statistical underpinnings of Figure 3C better explained in its figure legend to justify the claim that "Moreover, over 75% of these SNPs showed a coefficient of variation (CV) over 0.5" (line 146) in a clear manner? What does the bars, the boxes, and the central black line relate to?

9) Please replace "thosefrom" with "those from" (line 164).

10) From the sentence "We built a data table to store PMIDs linked to articles that contain potential drugs and associated risk alleles with ethnicity-specific information" (line 208) is not clear where the readers can access/view this data?

11) "pharmacogenomic" could be shortened to "PGx" (line 215).

12) Please change ">=0.4" to "≥0.4" (line 248).

13) Please replace "usedto" to "used to" (line 267).

14) It may not be directly clear what the authors mean by "Clinvar" (line 277)?

Author Response

First of all, we want to thank you for the constructive comments. The manuscript has been revised to address the reviewers’ concerns. Please see our point-to-point response below.

Reviewer 3:

Li et al. have come up with a very novel approach aimed at deciphering the complex landscape of patterns involved in pharmacogenomic (PGx) interactions. The authors have deployed data mining techniques to show that, despite the ethnical-specific clinical data are being scarce, indeed, conserved PGx patterns can be found between individual race groups. They have summarized these findings in a comprehensive database, including references to the the relevant literature sources, that should be publicly shared with the scientific audience. This study, the manuscript of which is extremely well written, provides one of the key cornerstones for us to enter the future era of personalized medicine.

1) Would it be possible to stratify the drug-SNP interactions further to incorporate even more personalized criteria such as the sex or age of the patient for each ethnic group? This data could be plotted in a new figure.

Thanks for the great suggestion. We are very interested to collect and explore such personalized patient information. However, such information is usually not reported in drug labels. We plan to add this to the literature research.

2) Please change "Correspondence:Corresponding author:" to "Corresponding author:" or "Correspondence:" (line 12).

Thanks for the comment. It is corrected.

3) "pharmacogenomics" could be shortened to "PGx" (line 57).

Thanks for the suggestion. It is adopted.

4) "Pharmacogenomic" could be shortened to "PGx" (line 58).

Thanks for the suggestion. It is adopted.

5) Although the authors state that the PGx ethnic database will be provided to researchers "Finally, all data was integrated into a database to provide researchers with information regarding drugs, paired PGx biomarkers, and associated allele frequency of SNPs for minority ethnic groups and to adjust drug doses and indications during drug development and drug application" (line 69), it seems not to be publicly accessible "The database will be shared upon request" (line 222). Please either stress this fact in the former sentence or make the database open to everyone.

Thanks for the suggestion. We revised the sentence and it reads “The Access database is published as Supplementary File 1 and future updated versions will be shared upon request.”

6) Would the authors mind providing a brief walk-through for Figure 1 in its legend? Please describe the process associated with each arrow. For example, it is not immediately clear what does "AF" mean and why does it connect to "Drugs" and "Biomarkers (genes)"?

Thank you very much for the suggestion, the detailed legend has been added for Figure 1.

It reads “Figure 1. Overall data collection workflow. The paired PGx biomarkers and drugs were first downloaded from TPGxBMDL. The oncology drugs were excluded based on the therapeutic area information within TPGxBMDL. Then, SNPs that potentially interacted with the remaining drugs were collected from public drug efficacy databases. These SNPs thus linked the drugs and the PGx biomarkers (genes) in which they located. Next, the allele frequency (AF) information of the SNPs across ethnic groups were collected from the ALFA database. Additionally, PubMed literatures were searched for potential information on ethnic minority groups regarding the PGx biomarker and drug pairs.”

7) Would it please be possible to have the 4 genes excluded from the drug-biomarker data set specified in parentheses (line 82)?

We are truly sorry for the mistake. Only three fusion genes (PML-RARA, BCR-ABL1, and FIP1L1-PDGFRA) were excluded. They were specified in the revised manuscript.

8) Would it please be possible to have the statistical underpinnings of Figure 3C better explained in its figure legend to justify the claim that "Moreover, over 75% of these SNPs showed a coefficient of variation (CV) over 0.5" (line 146) in a clear manner? What does the bars, the boxes, and the central black line relate to?

We added the values of min, max and median CV values in the boxplot. Some description was added for Figure 3c.

Also, we changed the sentence to “Moreover, 79% of these SNPs (26 of 33) showed a coefficient of variation (CV) over 0.5” to make it more clear.

9) Please replace "thosefrom" with "those from" (line 164).

The typo has been corrected.

10) From the sentence "We built a data table to store PMIDs linked to articles that contain potential drugs and associated risk alleles with ethnicity-specific information" (line 208) is not clear where the readers can access/view this data?

We revised the sentence and it reads “The PMIDs linked to articles that contain potential drugs and associated risk alleles with ethnicity-specific information were also stored in the database.” It is a separate data table in the database, and this was mentioned in Section 2.4.

11) "pharmacogenomic" could be shortened to "PGx" (line 215).

Thanks for the suggestion. It is adopted.

12) Please change ">=0.4" to "≥0.4" (line 248).

Thanks for the comment. It is corrected.

13) Please replace "usedto" to "used to" (line 267).

Thanks for pointing out the mistake. It is corrected.

14) It may not be directly clear what the authors mean by "Clinvar" (line 277)?

A brief description of ClinVar and the citation have been added.

This manuscript is a resubmission of an earlier submission. The following is a list of the peer review reports and author responses from that submission.